# Influenza virus transcription and progeny production are poorly correlated in single cells

David J Bacsik[1,2], Bernadeta Dadonaite[1], Andrew Butler[1], Allison J Greaney[1,2], Nicholas S Heaton[3,4], Jesse D Bloom[1,5]*

[1]Basic Sciences Division and Computational Biology Program, Fred Hutchinson Cancer Center, Seattle, United States; [2]Department of Genome Sciences & Medical Scientist Training Program, University of Washington, Seattle, United States; [3]Department of Molecular Genetics and Microbiology, Duke University School of Medicine, Durham, United States; [4]Duke Human Vaccine Institute, Duke University School of Medicine, Durham, United States; [5]Howard Hughes Medical Institute, Chevy Chase, United States

*For correspondence:
jbloom@fredhutch.org

**Abstract** The ultimate success of a viral infection at the cellular level is determined by the number of progeny virions produced. However, most single-cell studies of infection quantify the expression of viral transcripts and proteins, rather than the amount of progeny virions released from infected cells. Here, we overcome this limitation by simultaneously measuring transcription and progeny production from single influenza virus-infected cells by embedding nucleotide barcodes in the viral genome. We find that viral transcription and progeny production are poorly correlated in single cells. The cells that transcribe the most viral mRNA do not produce the most viral progeny and often represent aberrant infections that fail to express the influenza NS gene. However, only some of the discrepancy between transcription and progeny production can be explained by viral gene absence or mutations: there is also a wide range of progeny production among cells infected by complete unmutated virions. Overall, our results show that viral transcription is a relatively poor predictor of an infected cell's contribution to the progeny population.

## eLife assessment

This **important** paper reports a novel, **compelling** method, based on barcoding viral genes and next-generation sequencing, to quantify both viral transcription levels and progeny virus production in influenza virus-infected cells at the single-cell level. The authors show that viral transcription and progeny virus production are unexpectedly poorly correlated, and that cells in which viral RNAs are transcribed at high levels are not necessarily those producing the most progeny virions. Because of its novelty, the study will be of interest to the broader virology community.

## Introduction

Many aspects of viral infection are heterogeneous when measured across single cells. Individual infected cells vary widely in transcription of viral genes (*Russell et al., 2018*; *Sun et al., 2020*; *Zanini et al., 2018*; *Drayman et al., 2019*), expression of viral proteins (*Drayman et al., 2019*; *Brooke et al., 2013*; *Zhu et al., 2009*), presence of viral mutations (*Russell et al., 2019*), replication of viral genomes (*Schulte and Andino, 2014*), and production of viral progeny (*Zhu et al., 2009*; *Schulte and Andino, 2014*; *Delbrück, 1945*; *Heldt et al., 2015*). However, it is unclear how variation in these

different aspects of infection is related within the same infected cells. For instance, to what degree does the extent of viral transcription in an infected cell determine the number of progeny virions the cell produces? The answer to this question remains elusive because the most common single-cell techniques (flow cytometry and single-cell RNA sequencing) measure the levels of proteins and transcripts, rather than the number of viral progeny produced.

Here, we develop a novel approach to simultaneously measure viral transcription, viral mutations, and viral progeny production in single cells infected with influenza virus. We find that progeny production is even more heterogeneous than viral transcription in single cells. The cells that express the most viral transcripts often do not generate any detectable viral progeny. Instead, cells with extremely high viral transcription often fail to express the NS gene and represent non-productive infections. Our findings emphasize that different aspects of viral infection are not always correlated at the single-cell level, and that many of the cells contributing large amounts of viral mRNA to bulk RNA sequencing studies do not appreciably contribute virions to the progeny population.

## Results

### Viral barcoding to measure transcription, progeny production, and viral genotype in single cells

To quantify the progeny virions released from single infected cells, we inserted random nucleotide barcodes (*Lauring and Andino, 2011*; *Amato et al., 2022*; *Varble et al., 2014*) into the influenza virus genome so that they are positioned near the 3′ end of the viral mRNAs (*Figure 1A*). Standard 3′-end single-cell sequencing of the mRNA in infected cells (*Russell et al., 2018*; *Sun et al., 2020*; *Russell et al., 2019*; *Wang et al., 2020*; *Steuerman et al., 2018*; *Cao et al., 2020*) captures the viral barcode sequence, along with host and viral transcripts, enabling determination of which barcoded virion(s) infected each cell (*Figure 1A*). We can sequence the viral barcodes on progeny virions released into the supernatant to quantify the relative number of physical progeny produced by each cell and sequence the viral barcodes in cells secondarily infected with an aliquot of the supernatant to quantify the relative number of infectious progeny produced by each cell. Additionally, we can reconstruct the genome of the virion that infected each cell by selectively amplifying viral genes from the single-cell cDNA library and performing long-read sequencing as described previously (*Russell et al., 2019*). This strategy enables simultaneous measurement of transcription, progeny production, and viral genotype in single cells.

### Creation of dual-barcoded virus library

To insert barcodes into influenza virus genes, we used a previously described approach to duplicate the packaging signals of the HA and NA genes to create sites where exogenous sequence can be added without disrupting viral genome packaging (*Heaton et al., 2013*; *Gao and Palese, 2009*). This approach allowed us to insert 16-nucleotide random barcodes near the 3′ end of the genes, downstream of the stop codon but upstream of the polyadenylation signal (*Figure 1B*). These barcodes were therefore present in both viral mRNAs and genomic RNAs (vRNAs), but did not modify the amino acid sequence of the viral protein. We refer to these viruses as 'dual-barcoded' as they have barcodes on two different genes. Dual barcodes provided duplicate measurements of progeny production from the same infected cell, which were averaged and normalized (see 'Materials and methods').

We engineered barcodes into the A/California/04/2009 (pdmH1N1) strain of influenza virus with the G155E cell-culture adaptation mutation (*Chen et al., 2010*). Viruses with barcoded HA and NA segments could be generated by reverse genetics, and in cell culture grew to titers comparable to unmodified viruses (*Figure 1C*). To confirm that the sequence of individual barcodes did not affect viral growth in cell culture, we generated virus libraries carrying a small pool of barcodes and verified that the virus titers and barcode frequencies were stable across three passages (*Figure 1—figure supplement 1*).

For our single-cell experiments, we generated libraries of virions with a high diversity of barcodes on the HA and NA genes. Our experiments utilized between 1500 and 4000 infecting virions. It was important that nearly every virion have a unique barcode when randomly sampled from the library. We used deep sequencing to verify that for both HA and NA, in a sample of 1500 barcodes from our

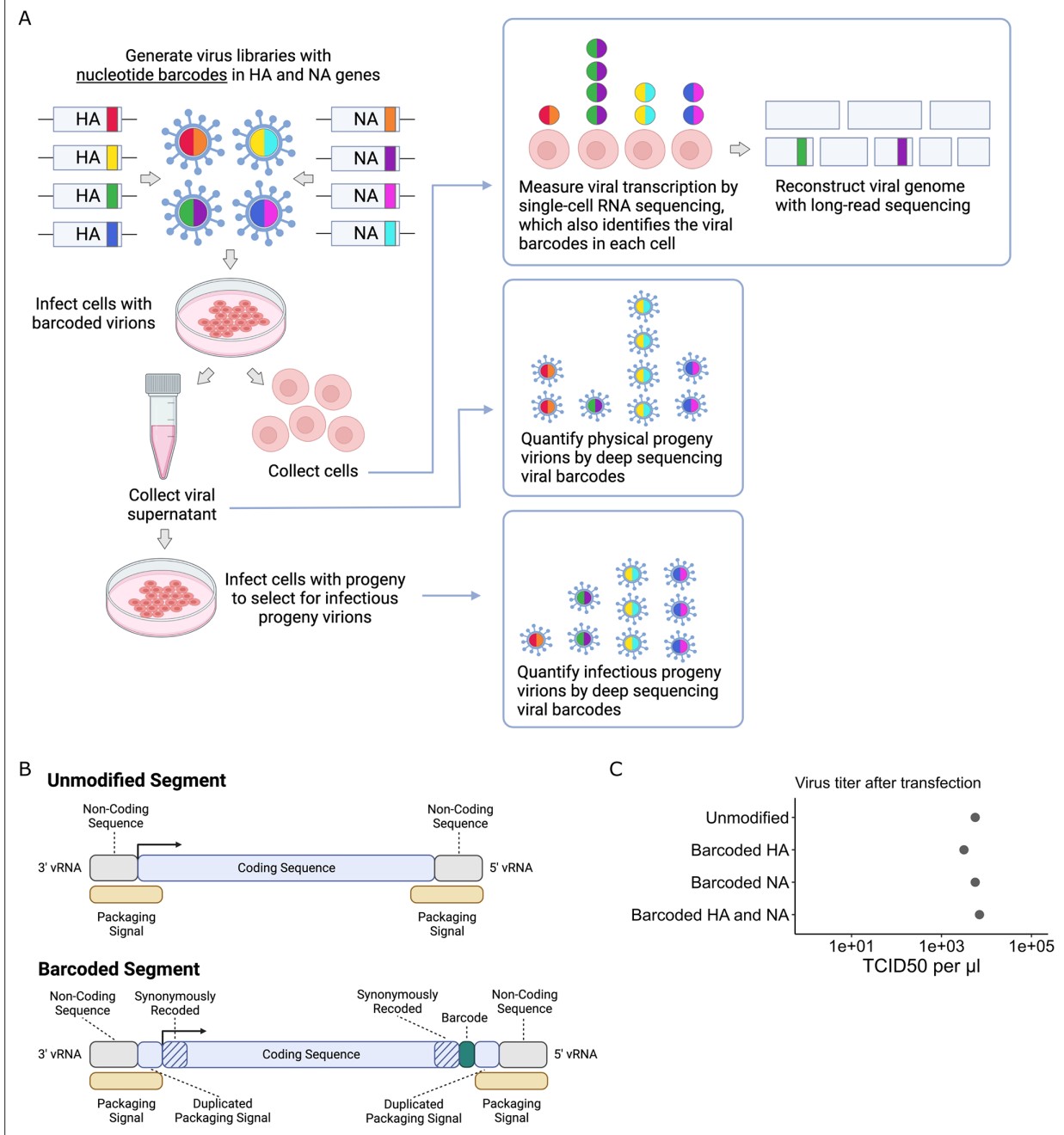

**Figure 1.** Strategy to measure transcription, progeny production, and viral genotype in single cells. (**A**) Insertion of barcodes in the viral genome makes it possible to quantify the progeny released from single cells and relate progeny production to viral transcription and viral genotype. (**B**) Barcodes were inserted near the 3' end of the mRNA sequence between the stop codon and the polyA site, using a duplicated packaging signal scheme to avoid disrupting viral genome packaging. (**C**) Viruses with one or two barcoded segments grew to similar titers as viruses with unmodified genomes. The titers shown were measured after generating the viruses by transfection.

The online version of this article includes the following figure supplement(s) for figure 1:

**Figure supplement 1.** Viral barcode sequences are selectively neutral.

**Figure supplement 2.** Extremely diverse barcoded virus libraries.

virus library, >96% of barcodes were unique (*Figure 1—figure supplement 2*). In a sample of 4000 barcodes, >92% were unique (*Figure 1—figure supplement 2*).

## We recapitulate prior findings that viral transcription is extremely heterogeneous across single infected cells

We implemented the experiment in *Figure 1A* at two different multiplicity of infections (MOIs). For the low MOI condition, we infected ~$10^4$ MDCK-SIAT1-TMPRSS2 cells (*Lee et al., 2018*) with the dual-barcoded virus library at an MOI of ~0.15; under these conditions, most cells are uninfected, and most infected cells are infected by a single virion. For the high MOI conditions, we infected ~$7 \times 10^3$ cells at an MOI of ~0.6; under these conditions, fewer cells are uninfected and a substantial fraction of the infected cells are infected by multiple virions. To ensure a single round of relatively synchronized infection, we replaced the virus inoculum with fresh medium after 1 hr and added ammonium chloride, which prevents secondary infection by blocking the endosomal acidification necessary for viral fusion (*Martin and Helenius, 1991*; *Ohkuma and Poole, 1978*). We collected the cells for single-cell RNA sequencing at a single timepoint 12 hr after infection. Before loading onto a 10X Chromium device, we added a control sample that allowed us to quantify the rate of cell multiplets and PCR strand exchange. This control sample contained cells infected with a second virus carrying synonymous mutations that could be distinguished in sequencing data (see 'Materials and methods').

For the low MOI sample, we obtained single-cell RNA sequencing data for 254 cells infected with our barcoded virus library, resulting in an empirical MOI of 0.17 (since we recovered data for ~1600 total cells from this sample). For the high MOI sample, we obtained single-cell RNA sequencing data for 357 cells infected with our barcoded virus library, resulting in an empirical MOI of ~0.59 (since we recovered data for ~800 total cells from this sample). Note that many cells are lost during preparation for single-cell RNA sequencing, so an important caveat of our study is that some infected cells that produced viral progeny are absent in the transcriptome data (*Zheng et al., 2017*; *Yamawaki et al., 2021*). However, because the number of barcodes in our virus library greatly exceeds the number of infected cells, these undetected infections should not substantially affect our measurements of relative viral progeny among the cells that were captured.

Under both infection conditions, there was extremely wide variation in the amount of viral transcription among infected cells (*Figure 2A*), similar to that observed in prior single-cell studies of influenza infection (*Russell et al., 2018*; *Sun et al., 2020*; *Russell et al., 2019*; *Wang et al., 2020*; *Steuerman et al., 2018*). In most infected cells, viral transcripts accounted for <10% of all transcripts, but in a small number of cells, over half of the transcripts were derived from virus (*Figure 2A*). On average, cells infected at high MOI expressed higher levels of viral transcripts than cells infected at low MOI, but there was extensive variation under both conditions (*Figure 2A*).

Under low MOI conditions, a substantial fraction (~40%) of the infected cells also failed to express all eight viral genes (*Figure 2B*), a phenomenon that has been extensively described in prior studies (*Russell et al., 2018*; *Sun et al., 2020*; *Brooke et al., 2013*). Under high MOI conditions, fewer cells failed to express all eight viral genes (*Figure 2B*), likely because of complementation by co-infection (*Phipps et al., 2020*; *Jacobs et al., 2019*; *Sims et al., 2022*). At a per gene level, each viral gene was not expressed in some infected cells, with variation between the rates of absence for each specific gene (*Figure 2C*). Note that our ability to determine whether a viral gene is absent depends on the total level of viral transcription in a cell (*Figure 2—figure supplement 1* and 'Materials and methods'), which could reduce our ability to detect the absence of the four viral genes involved in transcription (PB2, PB1, PA, NP).

## Full genome sequences of the influenza virions infecting individual cells at low MOI

Virions can be defective in two ways: they can fail to express a viral gene or they can encode mutated viral proteins. To identify cells infected by mutated virions, we used long-read sequencing to reconstruct the genome of virions infecting single cells under low MOI conditions (*Russell et al., 2019*). Under these conditions, we expect most cells that are infected to be infected by only one virion. We amplified the viral transcripts from our single-cell RNA sequencing library and subjected them to PacBio sequencing. Because each transcript carries a cell barcode, we could link the sequence of each viral transcript to the cell that produced it.

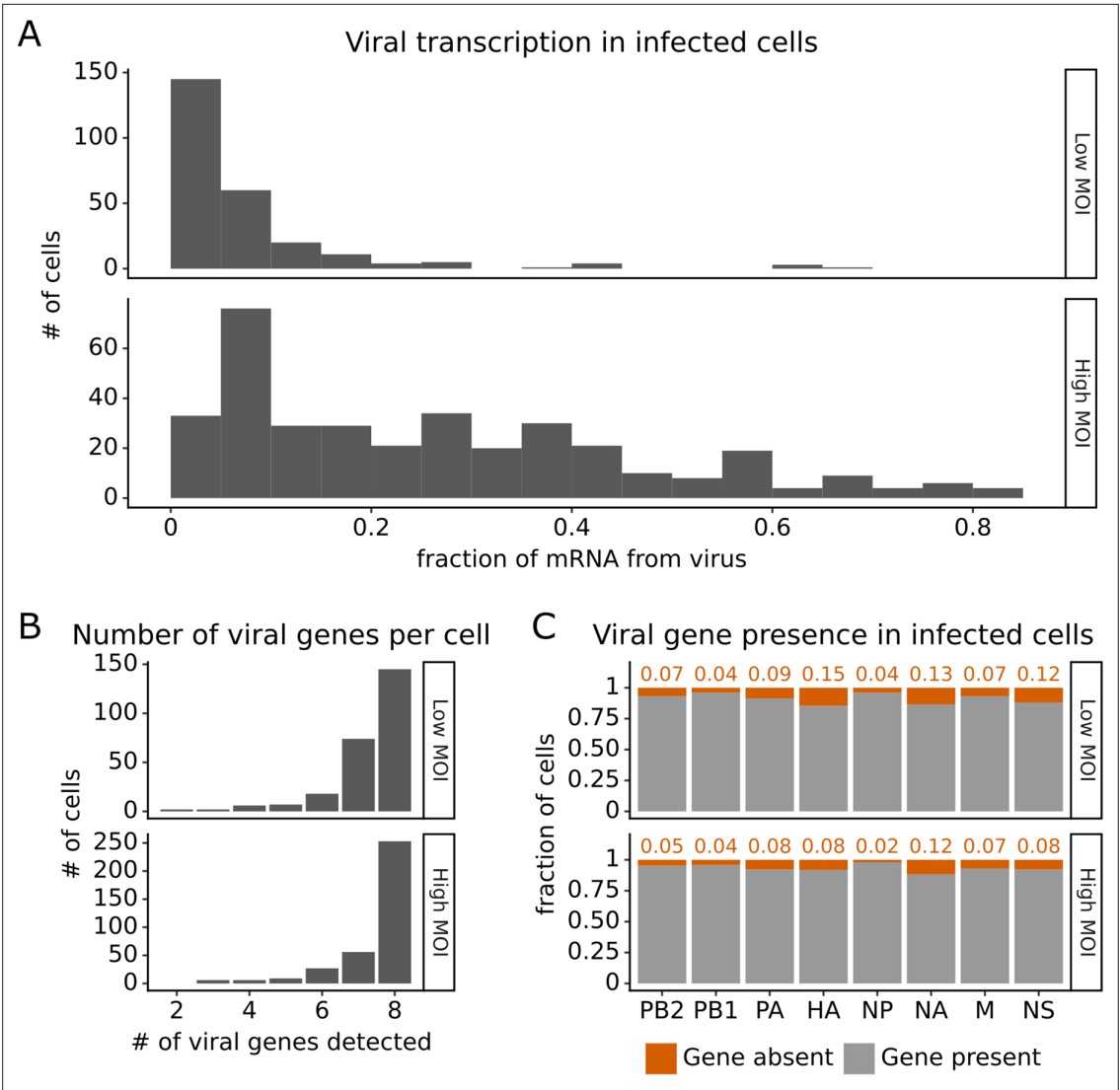

**Figure 2.** Viral transcription is extremely heterogeneous across single infected cells, and some cells fail to express some viral genes. This plot shows single-cell RNA sequencing data for the 254 cells that were infected at low multiplicity of infection (MOI) and 357 cells that were infected at high MOI. (**A**) Viral transcription in infected cells is extremely heterogeneous, with viral mRNA composing <1% of total mRNA in some cells, but >80% in others. (**B**) The number of viral genes detected in each infected cell. More than half of the infected cells express mRNA from all eight viral segments at both low MOI and high MOI. (**C**) The fraction of infected cells expressing each viral gene.

The online version of this article includes the following figure supplement(s) for figure 2:

**Figure supplement 1.** Expression of viral genes in infected cells.

We obtained complete sequences of all expressed viral genes for 131 of the 254 infected cells in our low MOI dataset (*Figure 3—figure supplement 1A*). About a third of the infected cells expressed all eight viral genes without any non-synonymous mutations (*Figure 3A*). The remainder of infected cells failed to express a viral gene, expressed a gene with a non-synonymous mutation, or both (*Figure 3A*). Mutated virions most commonly had just one non-synonymous mutation in their genome, but some virions had two or three mutations (*Figure 3B*). Note that some virions had large internal deletions in a gene (*Figure 3—figure supplement 2*) as has been previously described (*Saira et al., 2013*; *Davis et al., 1980*); here we have classified deletions as non-synonymous mutations.

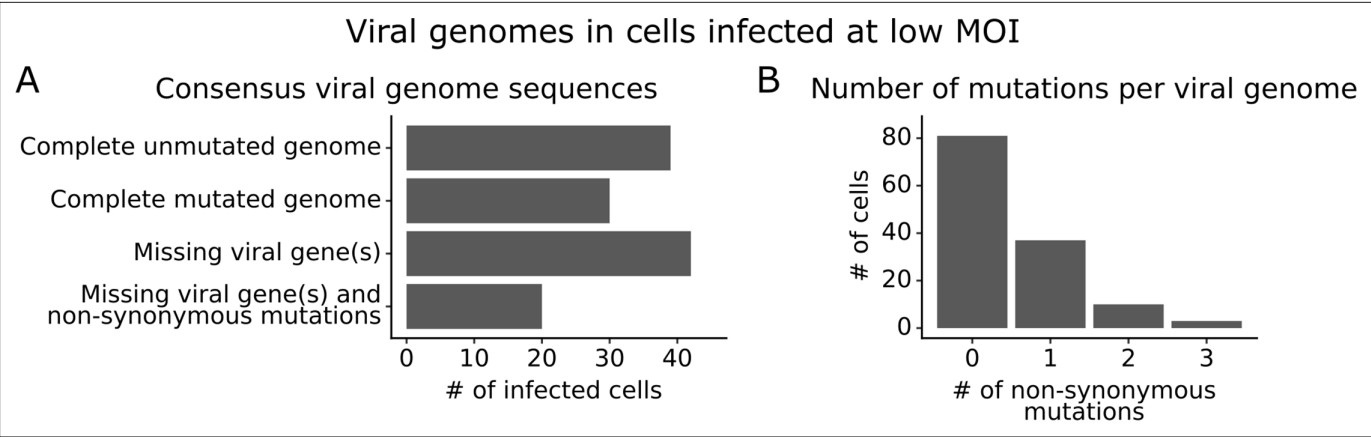

**Figure 3.** Consensus viral genome sequences from single infected cells with long-read viral sequencing data. Viral genomes were reconstructed for cells infected at low multiplicity of infection (MOI). (**A**) The number of single infected cells expressing all eight viral genes without non-synonymous mutations, expressing all eight viral genes with one or more non-synonymous mutation(s), missing one or more viral gene(s), or with both mutated and missing genes. (**B**) The number of non-synonymous mutations in each viral genome. Deletions are classified as a non-synonymous mutation for these counts. This plot shows only the 131 of 254 single infected cells for which we could determine the sequence of all genes expressed by the infecting virion. See *Figure 3—figure supplement 1* for details on properties of infected cells for which we could obtain full viral sequences, and *Figure 3—figure supplement 2* for the full set of viral mutations in each infected cell.

The online version of this article includes the following figure supplement(s) for figure 3:

**Figure supplement 1.** Number of cells with progeny measurements and viral genome sequencing.

**Figure supplement 2.** Viral genotypes in cells infected at low multiplicity of infection (MOI).

## Progeny production from single influenza-infected cells is more heterogeneous than viral transcription

We measured the amount of physical and infectious progeny virions produced by single infected cells. We quantified physical progeny virions by sequencing viral barcodes from vRNA molecules in the supernatant at 12 hr post-infection (*Figure 1A*). For the low MOI sample, we also quantified infectious progeny virions by infecting a second set of cells with some of the viral supernatant and sequencing viral barcodes from vRNA expressed in these newly infected cells (*Figure 1A*). We analyzed progeny production measurements for the 91 infected cells from the low MOI sample that met the following criteria: both barcoded genes were expressed (allowing us to identify both viral barcodes), and the sequences of all expressed viral genes were obtained with long-read sequencing (providing a complete viral genome) (*Figure 3—figure supplement 1*). For the high MOI sample, we analyzed progeny production measurements for the 290 infected cells that expressed both barcoded genes, since we did not have sequences of the viral genes.

The number of progeny virions produced per cell was extremely heterogeneous at both low MOI and high MOI (*Figure 4A and B*). Under low MOI conditions, nearly half of the infected cells failed to produce any detectable physical or infectious progeny. At the extreme high end of viral transcription, a few cells each generated >10% of all the virions detected in the progeny population (*Figure 4A and B*). A similar trend was seen at high MOI, although under these conditions the most productive cells generated a smaller fraction of progeny (*Figure 4A*).

Progeny production was much more heterogeneous across single cells than viral transcription (*Figure 4*). While just 6 of 91 infected cells were responsible for generating half of the physical progeny at low MOI, 23 cells were required to account for half of the viral transcripts (*Figure 4—figure supplement 1*). We can quantify the heterogeneity in these distributions formally by calculating a Gini coefficient, which ranges from 0 to 1, with larger values indicating more uneven distributions (*Gini, 1921*). Under low MOI conditions, the Gini coefficients were 0.78 and 0.88 for physical and infectious progeny production (*Figure 4A and B*). At high MOI, heterogeneity was reduced slightly and the Gini coefficient for physical progeny production was 0.66 (*Figure 4A*). The Gini coefficients for viral transcription were 0.46 at low MOI and 0.40 at high MOI (*Figure 4C*), both of which are lower than the corresponding coefficients for progeny production.

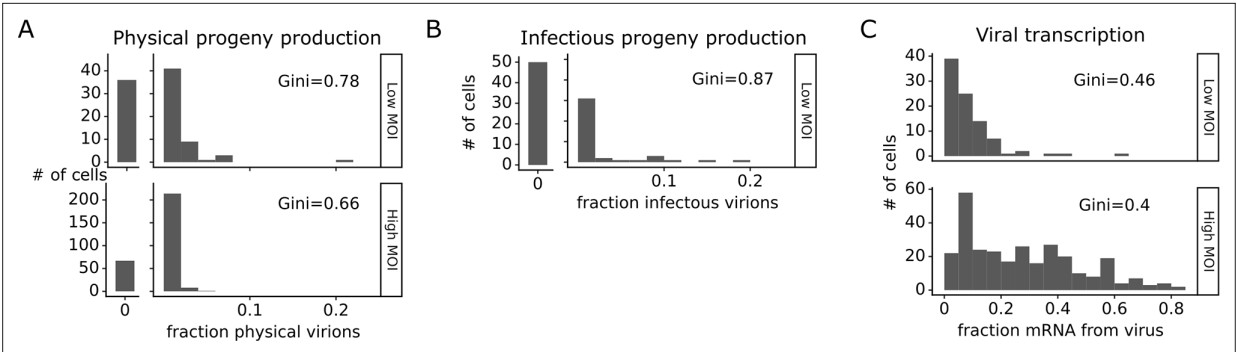

**Figure 4.** Viral progeny production is more heterogeneous than viral transcription across single infected cells. Heterogeneity across single infected cells in (**A**) physical progeny production, (**B**) infectious progeny production, and (**C**) viral transcription. The Gini coefficient (*Gini, 1921*) quantifying the extent of cell-to-cell variability is indicated on each panel; a larger Gini coefficient indicates a more uneven distribution. For (**A**) and (**B**), the x-axis is the fraction of viral barcodes associated with each cell among all barcodes assignable to any infected cell; for (**C**) the x-axis is the fraction of mRNA in each cell that is derived from virus. The outset bar on the left shows the number of cells that produced no detectable progeny. This plot shows only single infected cells with complete measurements (see *Figure 3—figure supplement 1*). For cells infected at low multiplicity of infection (MOI), these are cells that express both barcoded genes and for which we could determine the sequence of all genes expressed by the infecting virion. For cells infected at high MOI, these are cells that express both barcoded genes.

The online version of this article includes the following figure supplement(s) for figure 4:

**Figure supplement 1.** Cumulative fraction of viral products produced by single infected cells.

## Cells that transcribe more viral mRNA do not produce more progeny, and many high-transcribing cells represent aberrant infections that fail to express the NS gene

The correlation between viral transcription and progeny production in single cells is surprisingly poor at both low MOI and high MOI (*Figure 5A and B*). Under low MOI conditions, none of the cells with >25% of their mRNA transcripts derived from virus produce any detectable progeny (*Figure 5A*). Instead, most progeny come from cells with moderate viral transcription. At high MOI, viral progeny are produced by cells throughout the range of viral transcription (*Figure 5B*), but there is no trend for cells with high transcription to produce more progeny even at high MOI.

At both low and high MOI, the viral gene expression information provided by single-cell RNA sequencing offers a straightforward explanation for why some cells with very high viral transcription fail to produce progeny. Cells that fail to express any viral gene produce little or no detectable progeny virions, regardless of their total viral transcription activity. The lack of physical progeny produced by cells that fail to express even a single viral gene presumably occurs because the absence of the encoded protein impairs virion formation (*Figure 5D*; note that our analysis is limited to cells that express HA and NA since those are the barcoded genes). But although cells that fail to express a viral gene produce little or no progeny, the converse is not true: cells that express the full complement of viral genes often still fail to produce detectable progeny (*Figure 5D*).

Strikingly, absence of the influenza NS gene not only precludes progeny production but is specifically associated with an aberrant state of high viral transcription (*Figure 5C*). At both low MOI and high MOI, many of the highest transcribing cells fail to express NS, and the mean level of viral transcription is significantly higher (p<0.01) in cells that do not express NS compared to cells that express all viral genes (*Figure 5C*). These data suggest that NS acts as a negative regulator of viral transcription. This observation is consistent with the known functional roles of the NEP protein expressed from the NS gene, which is to export viral ribonucleoprotein complexes from the nucleus (*O'Neill et al., 1998*; *Bullido et al., 2001*) and possibly to switch the viral polymerase from transcription to genome replication through direct protein–protein interactions (*Robb et al., 2009*; *Mänz et al., 2012*). Overall, this result suggests that the cells that contribute the most to the signal observed in transcriptomic studies often represent aberrant non-productive infections that do not contribute viral progeny.

For cells infected at low MOI, we can also study the effect of viral mutations on progeny production. At low MOI, physical viral progeny are produced both by cells that express viral genes with mutations and by cells that express unmutated viral genes; however, more of the infectious progeny virions come

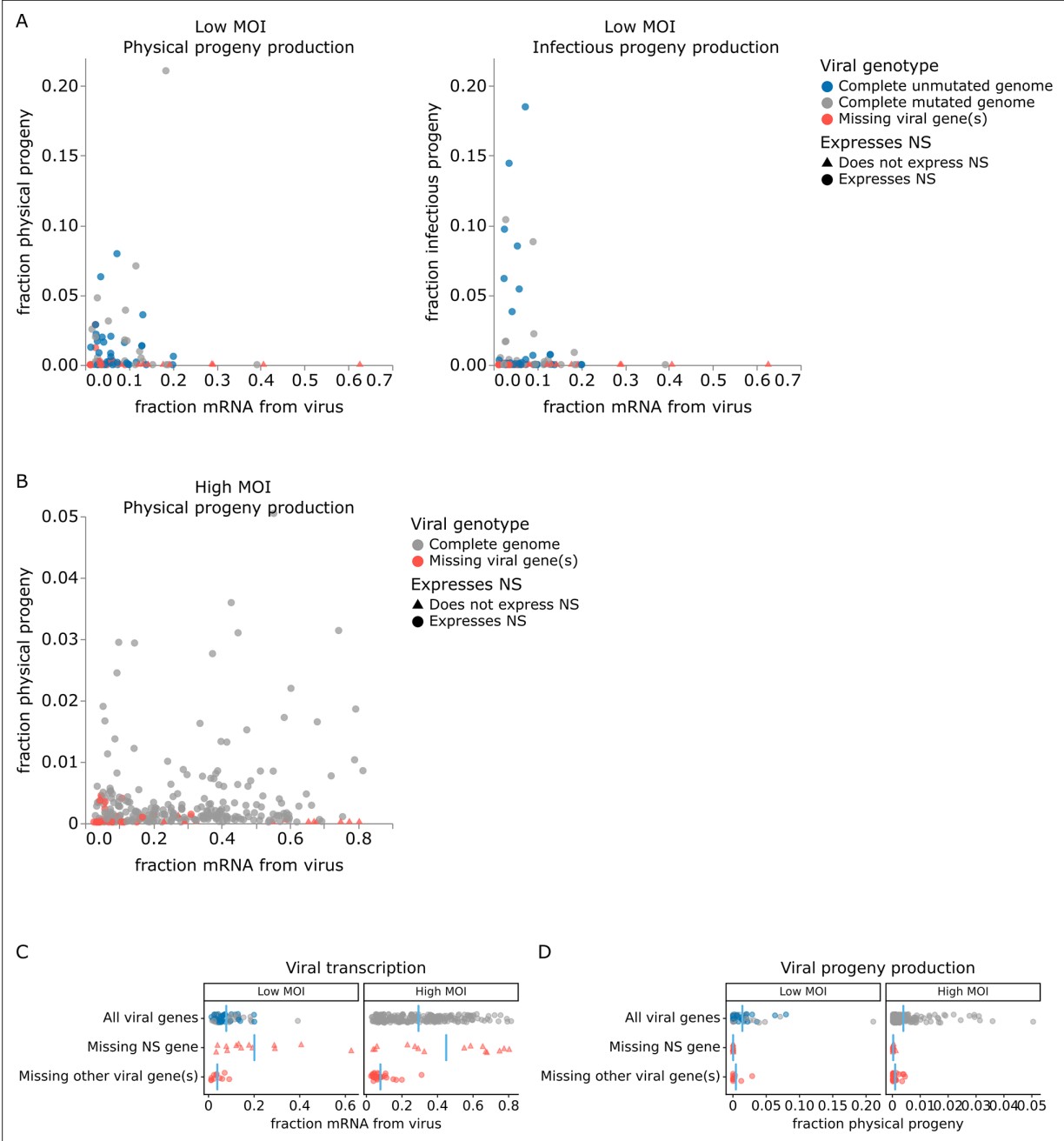

**Figure 5.** Relationship between viral transcription and progeny production in single infected cells. Relationship between viral transcription and progeny virion production. Each point is a different cell. (**A**) For cells infected at low multiplicity of infection (MOI), both physical and infectious progeny were quantified and cells are colored according to whether the cell expresses unmutated copies of all eight genes, all genes with one or more non-synonymous mutations, or fewer than all genes (with or without mutations). (**B**) For cells infected at high MOI, only physical progeny were quantified and cells are colored according to whether they express all eight viral genes or are missing one or more viral genes. Circular points indicate cells that express the NS gene, and triangular points indicate cells that do not express the NS gene. An interactive version of this figure that enables mouse-overs of points with details about individual cells is available at https://jbloomlab.github.io/barcoded_flu_pdmH1N1. (**C**) Total viral transcription is plotted for each cell. The mean for each group is shown as a blue line. Cells that do not express the NS gene transcribe significantly more viral mRNA than cells expressing all viral genes (statistical significance determined by permutation test with 5000 random simulations; p=0.0002 for cells infected at low MOI and p=0.004 for cells infected at high MOI). (**D**) Like panel (**C**), but for physical progeny production. Cells failing to express any viral gene – including NS – produce little or no physical progeny.

The online version of this article includes the following figure supplement(s) for figure 5:

**Figure supplement 1.** Frequency of physical progeny and infectious progeny from single infected cells infected at low multiplicity of infection (MOI).

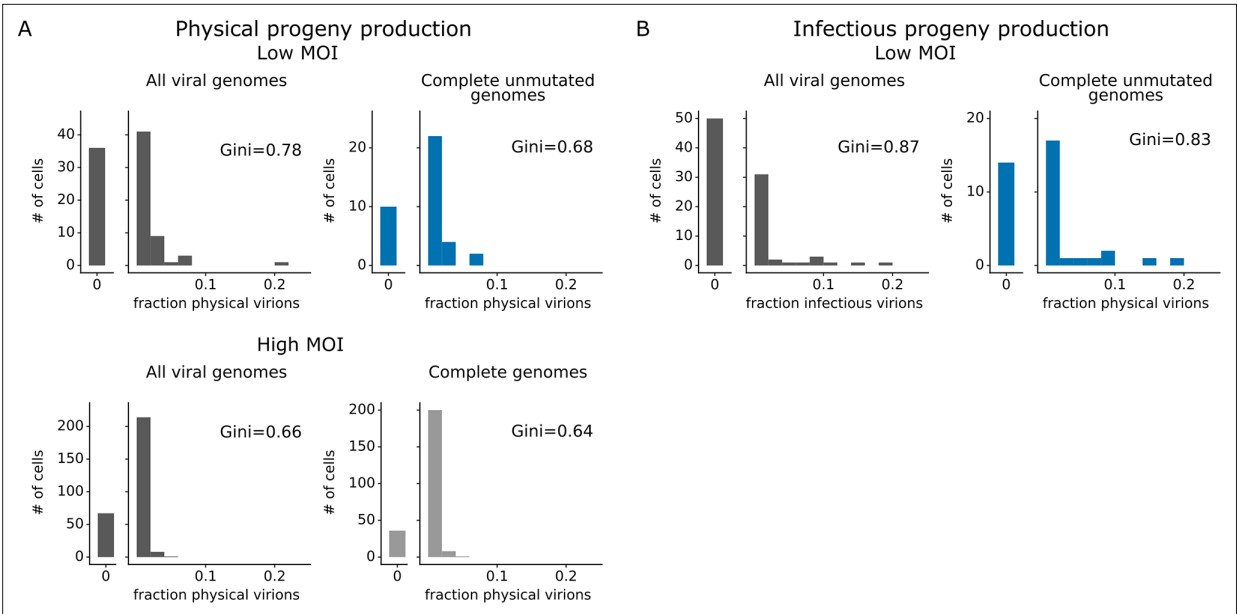

**Figure 6.** Viral gene absence and viral mutations only explain a fraction of heterogeneity observed in progeny production. (**A**) Distribution of physical progeny virions. For cells infected at low multiplicity of infection (MOI), compare heterogeneity in all infected cells (left, dark gray) with cells that express all eight viral genes without non-synonymous mutations (right, blue). For cells infected at high MOI, compare heterogeneity in all infected cells (left, dark gray) with cells that express all eight viral genes (right, light gray). The outset bar on the left shows the number of cells that produced no detectable progeny. (**B**) Like panel (A), but for infectious progeny rather than physical progeny at low MOI. The plots showing all infected cells are duplicated from *Figure 4* to facilitate direct comparison of all cells to those with complete unmutated genomes. For cells infected at low MOI, this figure shows the 91 infected cells for which we could identify the viral barcode on both barcoded genes and determine the sequence of all genes expressed by the infecting virion. For cells infected at high MOI, this figure shows the 290 infected cells for which we could identify the viral barcode on both barcoded genes.

The online version of this article includes the following figure supplement(s) for figure 6:

**Figure supplement 1.** Technical replicates of progeny measurements.

from cells with unmutated viral genomes (*Figure 5A*, *Figure 5—figure supplement 1*) – probably because some non-synonymous mutations interfere with protein functions that are required for infection of new cells. Nonetheless, physical and infectious progeny production are much more correlated among single cells than are transcription and progeny production (*Figure 5A* versus *Figure 5—figure supplement 1*; Pearson's R values of –0.14 for the correlation of transcription with infectious progeny at low MOI versus 0.39 for the correlation of physical progeny with infectious progeny at low MOI). However, none of the viral factors we measure (viral transcription, expression of each viral gene, or mutated viral proteins) fully explain the extreme variation we observe in progeny production. Progeny production remains highly variable across cells without any viral deficit (e.g. cells that express unmutated copies of all viral genes), although it is less variable across this population than across all cells (*Figure 6A and B*). This unexplained variation suggests that cellular or uncharacterized viral factors must also contribute to cell-to-cell variation in progeny production.

## Discussion

Most prior single-cell studies of infection have examined intracellular viral products, like mRNA transcripts or proteins (*Russell et al., 2018*; *Sun et al., 2020*; *Zanini et al., 2018*; *Drayman et al., 2019*; *Brooke et al., 2013*; *Zhu et al., 2009*; *Russell et al., 2019*). However, the most important outcome of infection for multi-cycle viral growth is how many progeny virions are produced by an infected cell. Prior studies of progeny production from single cells have relied on isolating individual infected cells in small volumes (*Zhu et al., 2009*; *Schulte and Andino, 2014*; *Delbrück, 1945*; *Heldt et al., 2015*). These studies have shown progeny production is highly heterogeneous, but have not provided measurements of most other properties of the infected cells – and have therefore lacked explanatory power to understand the basis for the variation in progeny production. Here, we have overcome these

limitations with a new method to simultaneously measure physical and infectious progeny production alongside transcription of all viral and host genes, as well as sequencing of the genome of the virion that infected each cell.

The most striking finding from our work is that while viral transcription and progeny production are both highly heterogeneous across influenza-infected cells, they are not well correlated at either low MOI or high MOI. In particular, the cells that transcribe the most viral mRNA often generate no detectable progeny virions. Part of the discrepancy between these two single-cell measurements is due to the fact that cells that fail to express the influenza NS gene tend to transcribe very high levels of mRNA but produce no progeny. This result makes biological sense: one of the proteins encoded by the NS gene is NEP, which exports viral ribonucleoproteins from the nucleus (*O'Neill et al., 1998*), terminating their transcriptional activity (*Bullido et al., 2001*). NEP may also mediate a switch from transcription to genome replication by the viral polymerase (*Robb et al., 2009*; *Mänz et al., 2012*).

However, absence of NS and other defects in the viral genome only explains part of the discordance between viral transcription and progeny production in single influenza-infected cells. These two properties are often discordant even in cells expressing unmutated copies of all influenza genes. We suggest that further study of both viral and cellular factors that promote transcription versus progeny production is an interesting area for future work.

Our study has several limitations. Our experiments used a cell line rather than the differentiated airway cells that are the actual target cells during human influenza infections. We performed experiments using a single strain of H1N1 influenza at a single timepoint. We were also able to profile all relevant single-cell properties for a relatively modest number of infected cells since single-cell RNA sequencing captures only a fraction of the input cells. However, the method we have described should be extensible to other cell-culture systems and larger numbers of cells in future work.

Despite these limitations, our results suggest several implications for broader thinking about viral infections. First, recent studies have examined the distribution of influenza virus transcription or protein expression across differentiated airway cells ex vivo (*Yamawaki et al., 2021*; *Kelly et al., 2020*; *Wang et al., 2020*) or in vivo (*Steuerman et al., 2018*; *Cao et al., 2020*; *Hamele et al., 2022*). Our results suggest it is also important to measure progeny production across airway cells as the cell types expressing the most viral transcripts or proteins may not be the ones producing the most viral progeny. Second, our results suggest failure of some cells to express specific viral genes contributes to the discordance between viral transcription and progeny production. Failure to express the influenza NS gene is strongly correlated with high viral transcription, even under high MOI conditions that promote genetic complementation (*Phipps et al., 2020*; *Jacobs et al., 2019*; *Sims et al., 2022*). Third, recent work on the transmission of influenza virus (*McCrone et al., 2018*; *Xue and Bloom, 2019*) and SARS-CoV-2 (*Braun et al., 2021*; *Martin and Koelle, 2021*; *Lythgoe et al., 2021*) in humans has emphasized the narrow genetic bottleneck, with only a small fraction of viral diversity in the donor transmitted to the recipient. Our results suggest physical bottlenecks in how many virions reach the recipient may be further narrowed by the fact that only a small fraction of the initially infected cells will produce most of the progeny that continue the infection.

## Materials and methods

The MDCK-SIAT1-TMPRSS2 were created and validated in the Bloom lab as described in *Lee et al., 2018*. The 293T (#CRL-3216) cells were obtained from ATCC. Both cell lines have been tested for mycoplasma contamination in our lab.

### Engineering barcodes in the influenza virus HA and NA genes

The HA segment of the A/California/04/2009 (pdmH1N1) strain of influenza virus with the G155E cell-culture adaptation mutation was engineered to carry exogenous sequence by duplicating the packaging signals at each end (*Heaton et al., 2013*; *Gao and Palese, 2009*), as schematized in *Figure 1B*. A complete plasmid map of the barcoded HA plasmid is available here. We included the G155E mutation as it greatly enhances viral growth in cell culture (*Chen et al., 2010*). Packaging signal length and location was informed by previous studies (*Gog et al., 2007*; *Watanabe et al., 2003*; *Marsh et al., 2007*). The terminal 105 nucleotides of the HA coding sequence were duplicated to provide an authentic packaging signal at the 5′ end of the vRNA. The corresponding 105 nucleotides of the HA

protein coding sequence were synonymously recoded to remove competing RNA-RNA interactions. A second stop codon (TGA) was added at the end of the coding sequence to reduce the chance of translation read-through. The stop codons were followed by an exogenous sequence containing a priming site, a 16-nucleotide random barcode, a second priming site, and a HindIII restriction site. The 3′ end of the vRNA was treated similarly. The first 67 nt of the HA coding sequence were duplicated, and the corresponding region of the coding sequence was synonymously recoded. All potential start codons were removed from the duplicated packaging signal using single-nucleotide substitutions. A BamHI restriction site was added between the duplicated packaging signal and the start codon.

The NA segment of the A/California/04/2009 strain was engineered using the same strategy except we duplicated 99 nucleotides at the 5′ end of the vRNA and 93 nucleotides at the 3′ end of the vRNA. A complete map of the barcoded NA plasmid is available here.

## Cloning barcoded plasmid libraries

To facilitate cloning highly diverse barcoded plasmid libraries, a recipient vector was created for each segment. The recipient vectors contained an eGFP insert flanked by the duplicated packaging signals described above. Recipient vector maps are available here.

Inserts were prepared by amplifying the HA and NA genes from templates with synonymously recoded terminal regions. Random barcodes were added as a string of 16 nucleotides in the primer that binds near the 3′ end of the viral mRNA. PCR was performed using KOD Hot Start Master Mix with 1 ng of plasmid template for 17 cycles. Reactions were treated with DpnI for 1 hr to remove the template plasmid. Barcoded products were gel purified and cleaned with 1X AmpureXP beads. The recipient vectors were prepared by digestion with BamHI and XbaI for 1 hr to remove the eGFP insert and linearize the backbone. Linear backbones were gel purified and cleaned with 1× AmpureXP beads.

Plasmids were assembled from linear vector and barcoded insert using NEBuilder HiFi Assembly Master Mix. A 2:1 molar ratio of insert to vector was used. 25 µl of NEBuilder Master Mix was combined with 0.27 pmol of barcoded insert and 0.13 pmol of linearized vector in a total volume of 50 µl. Assembly was allowed to proceed for 1 hr. Reactions were cleaned with 0.6× AmpureXP beads and eluted in 26 µl of EB. A small portion of the assembled product (1 µl) was used to transform 20 µl of NEB 10-Beta electrocompetent *Escherichia coli* cells. Transformation was performed at 1.8 kV for >5 ms per sample. Cells were grown in SOC media for 1 hr at 37°C with shaking.

After shaking, transformed *E. coli* were plated on large LB-ampicillin agar plates and grown at 37°C overnight to produce a 'lawn' of bacterial colonies. Liquid medium was pipetted onto the plate and a sterile plastic scraper was used to collect all of the bacterial colonies. Bacteria were grown in 200 ml of liquid medium in a 1 l flask for 4 hr at 37°C with shaking. Bacteria were pelleted by centrifugation and frozen at –20°C. Plasmid libraries were collected using QIAGEN HiSpeed Maxi Prep kit.

## Generating a dual-barcoded virus library

We generated a dual-barcoded virus library with all non-HA/NA genes derived from the A/California/04/2009 (pdmH1N1) strain of influenza virus. Virus was generated by reverse genetics in 39 independent transfection reactions. For each transfection reaction, 4e5 293T cells (ATCC #CRL-3216) were seeded in a well of a 6-well dish. Cells were grown in D10 medium (DMEM supplemented with 10% heat-inactivated fetal bovine serum, 2 mM L-glutamine, 100 U per ml penicillin, and 100 µg per ml streptomycin). After ~16 hr, we transfected each well with bidirectional reverse-genetics plasmids based on the pHW2000 vector (*Hoffmann et al., 2000*) carrying the six unmodified segments: (PB2, PB1, PA, NP, M, and NS), unidirectional reverse-genetics plasmids based on the pHH21 vector (*Neumann et al., 1999*) carrying the two barcoded segments (HA and NA), and a plasmid constitutively expressing the TMPRSS protease (which proteolytically activates HA) (*Lee et al., 2018*). Maps of all plasmids are available here. We used 250 ng of each plasmid and 3.4 µl BioT transfection reagent per reaction.

Twenty-four hours after transfection, the medium was replaced with Influenza Growth Medium (Opti-MEM supplemented with 0.1% heat-inactivated FBS, 0.3% bovine serum albumin, 100 µg per ml of calcium chloride, 100 U per ml penicillin, and 100 µg per ml streptomycin) and 3e5 MDCK-SIAT1-TMPRSS2 cells (*Lee et al., 2018*) were added to each well. Viral supernatants were collected at 65 hr

post-transfection and centrifuged at 500 RCF for 5 min to remove any cellular material. Aliquots were frozen at –80°C and titered by TCID50 assay.

To ensure a genotype–phenotype link between the viral genome and the proteins displayed on the surface of each virion, the virus library was passaged at low MOI. Infections were done at large scale to maintain library diversity. Four five-layer flasks (Falcon #353144) were seeded with 50 million MDCK-SIAT1-TMPRSS2 cells each (*Lee et al., 2018*) in D10 medium for a total of approximately 200 million cells. After 4 hr, the medium was removed and 2 million TCID50 units of virus library in IGM were used to infect the cells. Viral supernatants were collected at ~38 hr after infection and centrifuged at 500 RCF for 10 min to remove cellular material. Aliquots were frozen at –80°C and titered by TCID50 assay. We obtained titers of ~1e4 TCID50/µl (*Figure 1C*).

## Estimating the rate of infected cell multiplets and chimeric PCR products using a second control virus library

Immediately prior to performing single-cell RNA sequencing on our sample of interest, we mixed the infected cells with a second control sample of cells. The control cells were infected with an otherwise isogenic influenza virus that carried identifying synonymous mutations on all eight viral genes. The synonymous mutations are detectable by sequencing. They mark each mRNA transcript and genome segment derived from the virus library with a distinct 'genetic tag.' These synonymous genetic tags allow us to distinguish between viral transcripts from our sample of interest and the control sample, thereby enabling us to quantify two important sources of technical error.

First, in the single-cell RNA sequencing data, these tags provide a means to detect transcriptomes that are derived from droplets that encapsulated multiple infected cells (multiplets) (*Bloom, 2018*). Such transcriptomes are marked by high frequencies of both tags among the viral transcripts. The overall rate of multiplets among all cells was calculated, and multiplets bearing both tags (which will be about half of multiplets) were excluded to remove them from the dataset.

Second, the genetic tags are also detectable in the long-read viral sequencing data (*Russell et al., 2019*) we used to reconstruct the genotype of infecting virions. In the course of preparing long-read sequencing libraries, a polymerase can move from one template molecule to another in the midst of synthesizing its product – a phenomenon known as 'strand exchange' (*Judo et al., 1998*). This phenomenon can be detected in long-read viral sequences that contain discordant genetic tags (see Figure S10 of *Russell et al., 2019*). We estimated the rate at which this type of error occurs, and sequences bearing both tags were excluded from contributing to the results.

The second dual-barcoded virus library was prepared identically to the first viral library as described above. The second library contains synonymous variants near the 5' and 3' ends of each viral segment. Plasmid maps are available here.

## Infecting cells with a dual-barcoded virus library at low MOI

To infect cells at low MOI, $1 \times 10^4$ MDCK-SIAT1-TMPRSS2 (*Lee et al., 2018*) cells were suspended in D10 medium and plated in a well of 24-well plate. After 5 hr, cells were observed by microscopy and were confirmed to be well-attached. The medium was aspirated, and 1500 transcriptionally active units (measured by single-cell RNA sequencing) of dual-barcoded virus library in 100 µl of Influenza Growth Medium were added to the well. The cells were incubated with virus for 1 hr, and the plate was rocked by hand every 15 min. After 1 hr, the inoculum was removed and the cells were washed once with 250 µl of phosphate-buffered saline. Then, 500 µl of Influenza Growth Medium supplemented with 20 mM ammonium chloride (to prevent further entry of virions into cells *Martin and Helenius, 1991*; *Ohkuma and Poole, 1978*) was added to the well.

At 12 hr post-infection, the supernatant was collected and cells and debris were removed by centrifugation at 300 RCF for 3 min. The supernatant was split into two aliquots of 220 µl each and frozen at –80°C. The cells were collected by addition of 100 µl trypsin, and a single-cell suspension was generated. The trypsin digestion was stopped by addition of 400 µl of D10 medium. The cells were washed three times with phosphate-buffered saline supplemented with 0.8% by volume non-acetylated bovine serum albumin. The cells were counted to confirm that approximately 10,000 cells were present per well.

## Infecting cells with a dual-barcoded virus library at high MOI

High MOI infections were performed similarly to the low MOI infections described above. The following changes were employed: $6.7 \times 10^3$ MDCK-SIAT1-TMPRSS2 cells (*Lee et al., 2018*) were

plated and the cells were infected with 4000 transcriptionally active units (measured by single-cell RNA sequencing) of dual-barcoded virus library.

## Single-cell RNA sequencing

Infected cells were prepared and mixed with a second control sample of infected cells to control for technical sources of error (see 'Estimating the rate of infected cell multiplets and chimeric PCR products using a second control virus library' above). Approximately 20,000 cells (low MOI) or 13,000 cells (high MOI) were loaded into the 10X Chromium device. Single-cell RNA sequencing was performed with the 10X Chromium Next GEM Single Cell 3′ GEM, Library & Gel Bead Kit v3.1. The manufacturer's standard protocol (*Library Construction - Official 10x Genomics Support, 2022*) was used with the following modifications. The template-switching oligo was replaced with a modified single-stranded DNA oligo with the sequence 5′-AGAGTGTTTGGGTAGAGCAGCGTGTTGGCATGTrGrGrG-3′ at a final concentration of 45 µM in the reaction mix. This change was made to accommodate some of the barcoded influenza segments' exogenous sequence which shares homology with the standard 10× template-switching oligo. The cDNA amplification primer mix was replaced with a pair of primers with the sequences 5′-AGAGTGTTTGGGTAGAGCAGCG-3′ (binding to the custom template-switch oligo mentioned above) and 5′-CTACACGACGCTCTTCCGATCT-3′ (binding to the standard 10× adapter sequence) at a final concentration of 1 µM in the reaction mix. The cDNA amplification PCR reaction extension time was increased to 20 s to encourage the formation of full-length cDNA products. The amplified cDNA product was split in half. One half was used for fragmentation and preparation of the transcriptome sequencing library while, for the low MOI sample, the other half was used as template for long-read sequencing of viral transcripts.

## Viral long-read sequencing to reconstruct infecting viral genomes

We determined the sequence of the virion that infected each cell for the low MOI sample. Because the cells were infected at a low MOI, infection was initiated by one virion in the large majority of infected cells. To capture these sequences, we selectively enriched viral cDNA molecules using a method described previously (*Russell et al., 2019*). In brief, cDNA derived from the 10X Genomics protocol was first amplified in a semi-specific PCR reaction. Each segment was amplified with a primer annealing to the universal TruSeq primer site that is added to all cDNA molecules during the reverse transcription step of 10X Genomics protocol and a segment-specific primer annealing to 5′ end of the viral mRNA, which also contains a flanking sequence that is complementary to the TrueSeq primer site (*Supplementary file 1*). Semi-specific PCR reaction conditions were as follows: 12 ng cDNA, 0.5 µM of forward and reverse primer, 10 µl of KOD (EMD Millipore, 71842), 0.1 mg/ml BSA, and final volume adjusted to 20 µl with water. PCR was incubated for 120 s at 95°C, followed by 10 cycles of 120 s at 95°C, 20 s at 55°C, 90 s at 70°C, and the final extension step at 70°C for 120 s. Semi-specific PCR reactions were purified using AMPure XP beads at 1.8× beads to sample ratio and eluted in 12 µl of water. Following purification, PCR products were circularized via complementary TrueSeq sequence. For circularization, 10 µl of purified PCR product was used in a 20 µl HiFi assembly reaction (NEB, E2621S). HiFi assembly was performed at 50°C for 1 hr.

Next, HiFi products were used in segment-specific PCR reactions. To amplify viral products of all lengths, primers that anneal to the ends of viral mRNA were used; to preferentially amplify full-length viral segments, primers that anneal to the middle of each viral segment were used (*Supplementary file 2*). Segment-specific PCR conditions were as follows: 9 µl of Hifi reaction, 0.5 µM of forward and reverse primer, 25 µl of KOD, and the final PCR reaction volume adjusted to 50 µl with water. PCR was incubated for 120 s at 95°C, followed by cycling 120 s at 95°C, 20 s at 55°C, and 90 s at 70°C with a final extension step of at 70°C for 120 s. Cycles were kept to a minimum to reduce strand exchange; since different segments required different yield, different numbers of cycles were employed each segment-specific reaction. For the polymerase segments, 14 cycles of segment-specific PCR were performed;for the HA, NA, and NP segments, 10 cycles were performed; for the M and NS segments, 7 cycles were performed. PCR reactions were purified using AMPure XP beads at 1.8× beads to sample ratio and eluted in 12 µl of water. All purified PCR products were pooled together, and long-read sequencing was performed on a PacBio Sequel II.

We generated CCS sequences of each viral transcript using PacBio long-read sequencing. We measured the rate of strand exchange that occurred during sequencing library preparation (see Figure

S10 of *Russell et al., 2019*) and found that fewer than 1% of sequences were affected, providing high confidence that the sequences we obtained could be assigned to their cell of origin. We generated a consensus sequence for each viral genome (see 'Computational analysis of single-cell RNA sequencing, long-read virus sequencing, and progeny production viral barcode data' below). We counted the number of non-synonymous mutations found in each consensus genome; deletions were considered non-synonymous mutations for this purpose.

## Quantifying progeny production

The amount of progeny produced by single infected cells was determined by sequencing the viral barcodes on vRNA molecules. To quantify physical progeny virions, we sequenced the vRNA in the viral supernatant at 12 hr post infection. For the low MOI sample, to quantify infectious progeny virions, we infected a second set of cells to select for virions that could perform viral entry and genome replication (*Figure 1A*) and sequenced the intracellular vRNA molecules at 13 hr post infection.

In detail, we thawed frozen viral supernatants that were collected at 12 hr post infection and split them into four equal volumes. Two volumes were used to isolate supernatant RNA directly. For the low MOI sample, the other two volumes were used to infect MDCK-SIAT1-TMPRSS2 cells (*Lee et al., 2018*) at a moderate estimated MOI of ~0.25 in two independent replicates. To infect the cells, 60,000 MDCK-SIAT1-TMPRSS2 cells (*Lee et al., 2018*) were suspended in D10 medium and plated in a well of 6-well plate. After 7 hr, cells were observed by microscopy and were confirmed to be well-attached. The medium was aspirated and an aliquot of supernatant with an estimated 15,000 TCID50 units was added to the well in 500 µl of Influenza Growth Medium. The cells were incubated with virus for 1 hr, and the plate was rocked by hand every 15 min. After 1 hr, the inoculum was removed and the cells were washed once with 500 µl of phosphate-buffered saline. Then, 1600 µl of Influenza Growth Medium supplemented with 20 mM ammonium chloride (to prevent further entry of virions into cells *Martin and Helenius, 1991*; *Ohkuma and Poole, 1978*) was added to the well. At 13 hr post infection, the cells were collected by aspirating the growth medium and incubating with 300 µl trypsin to detach them from the plate. Trypsin digestion was stopped by the addition of 700 µl of D10 medium. The cells were pelleted by centrifugation at 400 RCF for 3 min. The cell pellet was washed by resuspending in 1 ml of phosphate-buffered saline and pelleting at 400 RCF for 3 min again. The phosphate-buffered saline was aspirated, and the cell pellet was flash-frozen on dry ice.

RNA was isolated from the viral supernatant or infected cell pellets using the RNeasy Mini Kit (QIAGEN, 74104). Lysis buffer was mixed with the viral supernatant sample and 70% ethanol was added. For the infected cell pellets, the sample was mixed with lysis buffer and homogenized by vortexing at high speed for 20 s. The homogenized sample was processed on a gDNA eliminator spin column to remove genomic DNA. The processed sample was combined with 70% ethanol. From this point, both the viral supernatant and infected cell pellets were treated identically and followed the standard RNA purification protocol specified by the manufacturer (*Qiagen, 2023*). The RNA for each sample was eluted in 50 µl of RNase-free water.

Reverse transcription was performed with a segment-specific primer targeted to the HA or NA vRNA (*Supplementary file 3*). Two replicate reactions were performed using RNA from the viral supernatant sample, and two independent reactions were performed using RNA from the two infected cell pellets; these replicates provide technical duplicate measurements of both the physical progeny in the supernatant and the infectious progeny in the cell pellets.

Reverse transcription was performed using the SuperScript III First-Strand Synthesis SuperMix kit according to the manufacturer's protocol (*ThermoFisher, 2023*). For the viral supernatant samples, 12 µl of each RNA sample was used as template for each 40 µl reaction. For the infected cell pellet samples which contain much larger amounts of total RNA due to the host RNA present in the cell, 1000 ng of RNA was used as template for each 40 µl reaction. The low-concentration cDNA generated from the viral supernatant samples was purified and concentrated using 2× Ampure SPRI beads and eluted into 22 µl of elution buffer.

Viral barcodes were amplified in 50 µl PCR reactions using KOD Hot-Start Master Mix (Sigma-Aldrich, 71842). For the viral supernatant samples, 22 µl of concentrated cDNA was used as template. For the high-concentration infected cell pellet samples, 10 µl of unpurified cDNA was used as template. Segment-specific primers (*Supplementary file 3*) were used and reactions were run for 20 cycles. Amplicons were size-selected and purified using a double-sided AmpureXP bead cleanup. Samples

were first combined with 0.8× AmpureXP beads, and the supernatant was collected. The supernatant was then combined with 1.8× AmpureXP beads, and the bound DNA was collected.

Sequencing indices and adapters were attached in a 50 µl PCR reaction using KOD Hot-Start Master Mix. For all samples, 2 ng of purified amplicon DNA was used as template. Sample-specific index primers (*Supplementary file 3*) were used and reactions were run for 20 cycles. The resulting amplicons were gel-purified and pooled for single-end sequencing on an Illumina MiSeq. The progeny contribution of each cell was calculated (see 'Computational analysis of single-cell RNA sequencing, long-read virus sequencing, and progeny production viral barcode data' below).

## Computational analysis of single-cell RNA sequencing, long-read virus sequencing, and progeny production viral barcode data

A reproducible pipeline that performs all analysis is available at https://github.com/jbloomlab/barcoded_flu_pdmH1N1. The pipeline uses Snakemake (*Köster and Rahmann, 2012*). The pipeline begins with raw sequencing data and ends by generating the figures shown in this article. Most code in the pipeline is arranged in Jupyter notebooks (https://jupyter.org).

Briefly, the raw sequencing data from the single-cell RNA sequencing was aligned using STARsolo RRID:SCR_021542 Version 2.7.6a (*Kaminow et al., 2021*) against a composite reference made up of the canine genome CanFam3.1.98 concatenated to the A/California/04/2009 influenza virus genome. Alignment produced a cell-gene matrix containing the gene expression of every canine and virus gene for each single cell. Custom Python code was used to parse the 'genetic tag' encoded on viral transcripts which differentiates our library of interest from a second control library. The multiplet rate was calculated, and only transcriptomes from our library of interest were used for analysis. Transcriptomes from the second control library and transcriptomes composed of multiple infected cells were excluded.

The total viral gene expression was calculated for each infected cell. Because the amount of viral transcripts in the ambient environment during single-cell RNA sequencing (*Young and Behjati, 2020*) varied by MOI, different thresholds were used to call cells as infected at each MOI. For the low MOI sample, cells were called as infected if at least 1% of their transcripts came from virus. For the high MOI sample, cells were called as infected if at least 2.5% of their transcripts came from virus. These thresholds provided clear separation of an infected and uninfected population. Individual viral genes were called as expressed if their frequency was greater than the 99th percentile observed in uninfected cells (see *Figure 2—figure supplement 1*).

For the statistical test in *Figure 5C*, we classified each cell as expressing all viral genes, missing the NS gene, or missing another influenza gene. Because the data are not normally distributed, we performed a non-parametric permutation test, randomly labeling each observed value with a classification. We performed 5000 permutations and calculated the p-value as the frequency with which the randomly generated difference between classified groups matched or exceeded the observed difference between groups. The results indicated that cells missing the NS gene have a statistically significant difference in viral transcription compared to cells that express all viral genes (p<0.01) under both low MOI and high MOI conditions.

For the low MOI sample, the raw PacBio sequencing data was processed using PacBio's ccs program RRID:SCR_021174 Version 5.0.0 (https://github.com/PacificBiosciences/ccs; *PacBio, 2022*). Consensus sequences were generated from the subread files, requiring a minimum accuracy (`rq`) of 0.99 for the consensus sequence. The chimera rate was estimated using the 'genetic tags.' The cell barcode and UMI were parsed from each CCS using custom Python code that utilized the alignparse package (*Crawford and Bloom, 2019*). A consensus sequence was called for each cell barcode–viral gene–UMI combination. A mutation was included in the consensus sequence if it was found in >50% of the CCS for the cell barcode–viral gene–UMI combination. A consensus sequence was then called for each cell barcode–viral gene combination. A mutation was included in the consensus sequence if it was found in >50% of the UMIs for the cell barcode–viral gene and was found in at least two UMIs.

To parse the viral barcodes sequenced from the supernatant (representing physical progeny) and from the second infection (representing infectious progeny), we used custom Python code that utilized the dms_variants package (https://jbloomlab.github.io/dms_variants/; *Bloom, 2021*). The viral barcodes were error-corrected using UMI-tools RRID:SCR_017048 Version 1.1.0 (*Smith et al., 2017*). The technical replicates for each sample were plotted against each other and the limit of detection

was set at 1e-5, where viral barcode frequencies fail to correlate (*Figure 6—figure supplement 1*), indicating bottlenecked subsampling of the molecules carrying the viral barcodes, and suggesting that frequency measurements below this threshold are not reliable; values below the limit of detection were set to the limit of detection. The mean frequency of both replicates was calculated. A subset of infected cells expressing both barcoded viral genes and with complete long-read sequencing data was used to calculate progeny contributions. To determine the fraction of progeny contributed by each infected cell in this set, we took the geometric mean of the HA and NA barcode frequencies associated with each cell. We normalized the progeny contributions by the total frequencies assignable to any cell in this set. The data were visualized in Jupyter notebook RRID:SCR_018416 Version 4.6.3 (https://github.com/jbloomlab/barcoded_flu_pdmH1N1/blob/main/final_analysis.py.ipynb). We used custom Python code utilizing a combination of plotnine (https://github.com/has2k1/plotnine; *Kibirige, 2023*; *Kibirige et al., 2020*) and altair (https://github.com/altair-viz/altair; *Altair, 2023*). An R script utilizing gggenes (https://github.com/wilkox/gggenes; *Wilkins, 2020*) was used to plot the complete viral genomes of infected cells. The figures generated by this notebook are displayed in this article.

## Acknowledgements

Thanks to Jason Underwood for providing valuable guidance related to performing long-read sequencing on single-cell cDNA libraries. We thank Will Hannon for assistance with interactive plots. BioRender was used to generate *Figure 1A and B*. This work was funded in part by the NIH/NIAID under grant R01AI165821 and contract no. 75N93021C00015, as well as using a Burroughs Wellcome Fund Young Investigator in the Pathogenesis of Infectious Diseases grant to JDB. JDB is an Investigator of the Howard Hughes Medical Institute.

## Additional information

### Competing interests

Allison J Greaney: Inventor on Fred Hutch licensed patents related to viral deep mutational scanning. The patent number is 62/692,398. Jesse D Bloom: consults or has recently consulted with Apriori Bio, Merck, Moderna, or Oncorus on topics related to viruses and their evolution. Inventor on Fred Hutch licensed patents related to viraldeep mutational scanning. The patent number is 62/692,398. The other authors declare that no competing interests exist.

### Funding

| Funder | Grant reference number | Author |
| --- | --- | --- |
| NIAID | R01AI165821 | Jesse D Bloom |
| NIAID | 75N93021C00015 | Jesse D Bloom |
| Burroughs Wellcome Fund | Young Investigator in the Pathogenesis of Infectious Diseases | Jesse D Bloom |
| HHMI | Investigator | Jesse D Bloom |

The funders had no role in study design, data collection and interpretation, or the decision to submit the work for publication.

### Author contributions

David J Bacsik, Conceptualization, Investigation, Visualization, Writing – original draft; Bernadeta Dadonaite, Investigation, Writing – review and editing; Andrew Butler, Software, Investigation, Writing – review and editing; Allison J Greaney, Nicholas S Heaton, Methodology, Writing – review and editing; Jesse D Bloom, Conceptualization, Supervision, Visualization, Writing – original draft

### Author ORCIDs

David J Bacsik http://orcid.org/0000-0003-4912-0209

Bernadeta Dadonaite ![ORCID] http://orcid.org/0000-0003-0908-6982
Andrew Butler ![ORCID] http://orcid.org/0000-0003-3608-0463
Jesse D Bloom ![ORCID] https://orcid.org/0000-0003-1267-3408

Public Review: https://doi.org/10.7554/eLife.86852.2.sa1
Author Response: https://doi.org/10.7554/eLife.86852.2.sa2

## Additional files

**Supplementary files**

• Supplementary file 1. Semi-specific primers for amplification of influenza transcripts from full-length cDNA library. These primers were used to amplify each influenza gene from the full-length single-cell RNA sequencing library. For each reaction, a segment-specific primer was paired with the 'Read1_TruSeq' primer, which binds to the Illumina sequencing primer found on all transcripts in the library.

• Supplementary file 2. Segment-specific primers for amplification of influenza transcripts. These primers were used to amplify the influenza genes from circularized templates. For each gene, two reactions were performed. One set of reactions targeted templates without large deletions and bound near the middle of the open reading frame; these reactions utilized the primers with 'mid' in their name. The other reactions targeted all templates (with and without deletions) and bound near the ends of the open reading frame; these reactions utilized the primers with 'end' in their name.

• Supplementary file 3. Primers for the reverse transcription, amplification, and indexing of viral barcode sequencing samples. Binding sequences are shown in uppercase and overhangs are shown in lowercase. Primers with 'RT' in their name were used to reverse transcribe the HA or NA viral RNA in supernatants or infected cells. The viral barcodes were amplified from the cDNA using an HA-specific or NA-specific primer (primers with 'PCR' in their name) paired with a primer that binds exogenous sequence embedded in the barcoded segments ('PCR_Universal_R'). The samples were prepared for pooling and Illumina sequencing by attaching a sample index ('SampleIndexXX_F') and sequencing adapters ('Adapter_Universal_R') in a final PCR reaction.

• MDAR checklist

**Data availability**

All data and code are available in the GitHub repository at https://github.com/jbloomlab/barcoded_flu_pdmH1N1, copy archived at *Bloom, 2023*. The analysis can be reproduced by running the Snakemake pipeline and final analysis notebook according to the instructions at https://github.com/jbloomlab/barcoded_flu_pdmH1N1/blob/main/README.md. Key output files are hosted at the following locations. All raw sequencing files are available on GEO under the accession number GSE214938. The single-cell RNA sequencing cell-gene matrix is also available on GEO under the accession number GSE214938. An integrated CSV produced by the Snakemake pipeline with cell barcodes, viral gene expression, viral genome sequence, and viral barcode frequencies is available at https://github.com/jbloomlab/barcoded_flu_pdmH1N1/blob/main/results/viral_fastq10x/all_samples.csv. The final CSV file with progeny contribution measurements, viral gene expression, and viral mutations (when available) for the infected cells with complete measurements is available at https://github.com/jbloomlab/barcoded_flu_pdmH1N1/blob/main/results/viral_fastq10x/all_samples_complete_measurements_cells_data.csv.

The following dataset was generated:

| Author(s) | Year | Dataset title | Dataset URL | Database and Identifier |
|---|---|---|---|---|
| Bacsik DJ, Dadonaite B, Butler A, Greaney AJ, Heaton NS, Bloom JD | 2022 | Influenza virus transcription and progeny production are poorly correlated in single cells | https://www.ncbi.nlm.nih.gov/geo/query/acc.cgi?acc=GSE214938 | NCBI Gene Expression Omnibus, GSE214938 |

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
