## [Editor Report · eLife assessment]

This **important** paper reports a novel, **compelling** method, based on barcoding viral genes and next-generation sequencing, to quantify both viral transcription levels and progeny virus production in influenza virus-infected cells at the single-cell level. The authors show that viral transcription and progeny virus production are unexpectedly poorly correlated, and that cells in which viral RNAs are transcribed at high levels are not necessarily those producing the most progeny virions. Because of its novelty, the study will be of interest to the broader virology community.

---

## [Referee Report · Public Review]

In this article, a novel technique allowing the linking of viral transcription levels and progeny virion production is presented. Barcoded libraries of an H1N1 influenza virus (two genes were barcoded near the 3'end) were used to infect cells using an experimental approach ensuring that, in the low multiplicity of infection condition, each cell is infected by one virion and that nearly every virion has a unique barcode. This allows then, upon single-cell RNA sequencing and sequencing of the supernatants, to infer back the cells that were producing certain barcoded viruses. Assessing detection frequencies of barcodes in the single-cell sequencing and in the sequencing of the supernatants allows us to compare the levels of viral transcription and progeny virion production.

Observations that viral transcription levels are very heterogenous at the single-cell level are not novel, but reinforce those from previous studies. The major findings of this study are (i) progeny virion production is also very heterogenous, i.e., a few cells produce most of the progeny virions and (ii) there is a poor correlation between viral transcription levels and progeny virion production at the single-cell level.

Strengths:

The article is very well written, the experimental choices are very well justified and the methods are very detailed, allowing the possibility of reproducing the work performed in this study. The conclusions are very well supported by the data and the limitations of the study and how those might influence the conclusions are also clearly explained. In addition, several experimental caveats, such as PCR cross-overs in next-generation sequencing and cell multiplets in single-cell sequencing, were well accounted for, which is not always the case in studies using these techniques.

Weaknesses:

It seems that the results presented here are from one single experiment. How reproducible are the results?

As explained in the article, it is important that nearly every virion has a unique barcode. This was assessed by sequencing the barcodes in the virus libraries. Between 92% to 96% of the barcodes were unique. With this information, it should be possible to assess whether non-unique barcodes were detected in infected cells, and if yes, remove these from the downstream analysis.

It seems like all the information available in this very rich dataset was not fully exploited. For instance, Figure 5C suggests that cells missing the expression of one viral gene might still be able to produce progeny viruses. It would be interesting to have the information regarding which gene was not expressed in these cells.

The introduction and discussion are rather short and the article could benefit from expanding them. Additional speculations about viral or cellular factors (e.g. differences in innate immune responses, differences in cell division status) that might explain the observed heterogeneity, both at the viral transcription and viral progeny virus production levels, would be interesting.

---

## [Author Response]

We thank eLife and the reviewer for the nice summary of our manuscript. We largely agree with the summary and review, and just add a few small points.

First, the review asks about the reproducibility of our findings, and suggests that they are only from a single experiment. In fact, our manuscript reports data from two independent single-cell experiments: one performed at low multiplicity of infection (MOI), and another at higher MOI. The broad trends, including the lack of strong correlations between viral mRNA transcription and progeny production, are consistent across both experiments.

Second, the reviewer asks about what happens when two different virions bearing the same viral barcode infect two different cells, given that we estimate 4-8% of barcodes to be shared between multiple infecting virions. When two cells are infected by different virions with the same barcode, this breaks the one-to-one link between transcription in that cell and progeny in the supernatant, since it is not possible to determine which cell contributed the progeny with that barcode. This means that between 4-8% of the points on our correlation plots could be affected by this factor, meaning that a few outliers should be expected. Another scenario, where a single cell is infected by two barcodes, is not problematic for our method because we can simply sum the progeny output for both barcodes from that cell.

Finally, the reviewer notes that some cells appear to produce progeny virions despite failing to express one or more viral genes. Such cells can be explained in one of two ways. First, as noted immediately above, we expect a small fraction (4-8%) of the points to be erroneous due to a lack of a guaranteed one-to-one link between cell and progeny for non-unique barcodes. Second, in some cases the missing viral gene could be a technical artifact caused by a stochastic failure to capture modestly expressed transcripts from the gene; this phenomenon, known as gene dropout, occurs at a fairly high rate in single-cell experiments (see Qiu Nature Communications 2020 for a detailed discussion). Genes that are expressed at lower levels, like the Influenza virus polymerase genes, are more likely to be missed during single-cell RNA sequencing. The absent viral genes in each infected cell can be explored in detail using the interactive plots at https://jbloomlab.github.io/barcoded_flu_pdmH1N1/